# Environmental Tobacco Smoke and Early Language Difficulties among U.S. Children

**DOI:** 10.3390/ijerph18126489

**Published:** 2021-06-16

**Authors:** Dylan B. Jackson, Alexander Testa

**Affiliations:** 1Johns Hopkins Bloomberg School of Public Health, Johns Hopkins University, Baltimore, MD 21205, USA; 2Department of Criminology & Criminal Justice, University of Texas at San Antonio, San Antonio, TX 78249, USA; alexander.testa@utsa.edu

**Keywords:** smoking, environmental tobacco smoke, language, children, receptive, expressive

## Abstract

**Objective**: Environmental Tobacco Smoke (ETS) is a serious public health concern with the potential to interfere with various components of healthy child development. Even so, there has been limited nationally representative research investigating these connections. The current study examines the relationship between ETS and language difficulties among toddlers and preschool-aged children in the United States. **Method**: Data are derived from the 2018 National Survey of Children’s Health and facilitate strategic comparisons between different forms of ETS—namely, children who live with family members who smoke vs. children whose family members smoke inside the housing unit. **Results**: The findings reveal a robust association between family members smoking inside the housing unit and both receptive and expressive language difficulties, but only among male children. After adjusting for covariates, smoking inside the housing unit is associated with a 182% increase in the rate of early composite language difficulties among male children. These associations persist even when compared to male children who live with smoking family members who do not smoke inside the housing unit. **Conclusions**: The findings suggest a need for interventions designed to reduce ETS in households with young children and increase targeted language skill training for vulnerable children in an effort to enhance child development and well-being. To maximize this effort, we advocate for interdisciplinary teams, including medical and public health practitioners, educators, and researchers, to work together to develop and implement evidence-based strategies to limit ETS in homes and facilitate healthy language development among young children.

## 1. Introduction

Tobacco smoke exposure is one of the leading causes of preventable disease and premature mortality in the United States and worldwide [1,2]. Estimates suggest that approximately one in four non-smokers (approximately 58 million persons) in the United States were exposed to secondhand smoke during 2013–2014, with the exposure prevalence being highest among children ages 3–11 years old [3]. It is estimated that between 1964 to 2014, over 2.5 million Americans lost their lives as a result of secondhand smoke (SHS) exposure [4]. Aside from the toll on human life, SHS exposure also generates a substantial economic cost of approximately $5.6 billion per year in lost work productivity alone [4]. Given the substantial cost to health and economic well-being, the United States Surgeon General proclaimed that there is no acceptable risk-free level of SHS exposure more than a decade ago [5].

One of the most potent risk factors for SHS exposure (i.e., inhalation of environmental tobacco smoke) is the actual presence of environmental tobacco smoke (ETS) in a given setting, which underpins efforts of states and localities in recent years to implement bans on smoking in public places [6]. However, when it comes to young children, research has indicated that parent/caregiver smoking—particularly indoors—is one of the most potent risk factors for child SHS exposure [5,7]. Despite some promising trends in the prevalence of smoke-free rules in homes with children [8], many children still encounter indoor smoking, with exposure “occur[ing] predominantly through smoking by caregivers in the family home” [9]. The presence of ETS in a child’s home is a serious concern given its connection to SHS exposure, which in turn increases the risk of multiple illnesses (e.g., lower respiratory infections, otitis media, asthma) and premature mortality among children [4,10,11].

A growing body of research, moreover, has revealed that the negative downstream impact of ETS likely extends beyond the physical health of the children to other facets of healthy development, such as those pertaining to mental health, behavior, and emotions. For instance, a recent study of U.S. children living in areas of rural poverty revealed a significant, linear association between ETS and children’s symptoms of hyperactivity and conduct problems [12]. Another study by Sevcikova and colleagues [13] employed a sample of 1478 school children in Slovakia and found that, while ETS was associated with lower maternal education and family SES, it was also associated with child emotional and behavioral difficulties. These associations were found whether the source of the ETS was the mother or the father smoking inside the housing unit. Finally, a recent study by Mahabee-Gittens and colleagues [14] of a nationally representative sample of 6–11-year-old children revealed that those who lived in homes with ETS were more likely to exhibit a variety of mental health and neurodevelopmental conditions, including attention deficit hyperactivity disorder (ADHD), depression, and learning disabilities. The authors note that many of the associations between ETS and child outcomes were more robust in younger children whose parents reported smoking inside the housing unit. It is also worth noting that, to the extent that ETS increases the risk of SHS exposure among children, it may further elevate the risk of a variety of neurobehavioral, cognitive, and mental health challenges [15,16,17,18,19].

Despite evidence linking ETS to a variety of deleterious outcomes among children, there has been limited research investigating whether ETS is associated with language difficulties during the first years of life. Early childhood is a critical developmental stage for the formation of language skills, given the proliferation of neural pathways during this stage that support language development [20,21]. Notably, early language difficulties have been shown to elevate the risk of various deleterious outcomes across the life course [22]. Specifically, early language skills have been linked to key markers of subsequent health and development, including academic achievement [23], literacy skills [24,25], and emotional and behavioral outcomes [26,27]. There has also been growing attention to social factors that explain disparities in language skills among children [26,28], with several public awareness and action campaigns designed to elevate public consciousness concerning the importance of enhancing children’s language skills and expanding access to high-quality environments that support language skill development among vulnerable children [29,30,31]. It is possible, however, that existing campaigns may be overlooking ETS as a robust risk factor for language difficulties during the early life course that may need to be considered alongside other important family characteristics and features of the environment.

Notwithstanding this possibility, limited research has investigated how exposure to ETS—particularly during infancy and early childhood—may be harmful to language development. A study by Yolton and colleagues [32] found a significant, inverse relationship between ETS exposure and reading scores. Even so, this research employed a sample of older children and youth (ages 6 to 16) that was obtained between 1988 to 1994. However, more recent research by Oh and colleagues [33] pointed to the possibility of a connection between ETS exposure during toddlerhood and early childhood and language difficulties. Examining a sample of mothers and children in Cincinnati, Ohio, they found that tobacco exposure at ages 1, 2, 3, and 4 was significantly and negatively associated with middle-childhood executive functions, which have been shown to exhibit reciprocal associations with language skills [34]. Notwithstanding these findings, there is a dearth of nationally representative research on the impact of early life ETS exposure on language difficulties—particularly among toddlers and preschool-aged children. This is a notable limitation, given findings from previous studies showing that ETS can inhibit proper development across a variety of domains (e.g., mental health, behavior), as well as growing interest in understanding disparities in childhood language abilities. As Bornstein and colleagues [22] recently noted, “language is among the most complex skills a child must master…so understanding the individual differences in language and their developmental stability is of compelling interest to professionals, practitioners, and parents”.

Furthermore, there are established sex differences in language development among children [35,36], with boys appearing more vulnerable to aversive environments [37,38]. Consequently, it is possible that the link between ETS exposure and early language difficulties may be concentrated among male children. To elaborate, not only are there early sex differences in language development that typically disadvantage boys [35], but the neurodevelopment of boys may be significantly more susceptible to an environmental hazard, such as ETS. Several studies to date have revealed male neurodevelopmental vulnerability to other environmental hazards. For instance, research has shown that, relative to female children, male children exhibit more significant cognitive impacts following early exposure to other toxins, such as lead [39,40], as well as air pollutants, including particulate matter and nitrogen dioxide [41]. In these cases, researchers have hypothesized that oxidative stress and systemic inflammation may be the process by which exposure to these toxins might differentially impact neurodevelopment in male and female children, producing a stronger effect in boys [41]. Additionally, there may be several physiological and chemical differences between boys and girls in relation to neurodevelopment that extend to diverse learning and cognitive phenotypes [42]. While this research largely focuses on exposure during the prenatal stage, recent studies suggest that even postnatally, early childhood exposure to toxins has significant potential to undermine children’s neurodevelopment, especially among boys [43,44]. Still, research has yet to explore whether such is the case for the language development of boys exposed to ETS.

The current study aims to increase knowledge regarding the relationship between ETS and early language skill development among young male and female children in the United States. Using recent, nationally representative data from the 2018 National Survey of Children’s Health (NSCH), we examine the association between ETS exposure among children (ages 1–5 years) and early language difficulties. Furthermore, given the research on male neurodevelopmental vulnerability to environmental toxins, we also examine these associations using sex-stratified models.

## 2. Materials and Methods

### 2.1. Data

The NSCH is a survey of a cross-sectional probability sample of U.S. children, ranging in age from 0 to 17 years. The survey is funded by HRSA’s Maternal and Child Health Bureau and conducted by the U.S. Census Bureau. The sample was taken from the Census Bureau’s Master Address File, which contains a complete listing of all known residences in the U.S. and the District of Columbia, and includes an administrative flag to identify households that are most likely to have children [45]. The survey assesses multiple intersecting components of children’s lives and includes items that ask primary caregivers of focal children about the health and well-being of children as well as their development of children across a variety of domains (i.e., behavioral, social, cognitive, etc.). In the 2018 survey, caregivers are also asked about the early language development of 1 to 5-year-old children and whether the focal child lives in a household with a greater risk of exposure to ETS, making the data well-suited to the present study.

Despite the existence of previous iterations of the survey, only the 2018 NSCH data was used in the present study, as it is the most recent year available, and previous cohorts do not include the present items pertaining to early language development of 1–5-year-old children. A total of 30,530 surveys were completed in 2018. However, given the focus of the present study, we restricted the sample to households with focal children ages 1 to 5 years (*N* = 7594), as these are the only households who were presented with the early language development items. In order to address missing observations in multivariate analyses, we present multiply imputed results calculated in STATA 16.1 using the MI commands (chained equations; 20 imputations). The data employed for the present study are publicly available through the Child and Adolescent Health Measurement Initiative and therefore did not require IRB approval.

### 2.2. Dependent Variable

The 2018 NSCH survey was the first of the NSCH surveys to include 11 items pertaining to early language development in a section entitled, “This Child’s Learning”. These items were developed by HRSA’s Maternal and Child Health Bureau as part of a broader initiative to develop national measures of the extent to which children in the U.S. are “Healthy and Ready to Learn” [46,47]. These efforts are further supported by the Data Resource Center for Child and Adolescent Health—a project of the Child and Adolescent Health Measurement Initiative (CAHMI) (https://www.cahmi.org/) (accessed on 20 November 2020).

*Receptive.* Five of the 11 items pertaining to early language development assess children’s receptive language skills. Specifically, primary caregivers were asked whether the child was able to do the following at the time of the survey: (1) Understand the meaning of the word “no”, (2) Follow a verbal direction without hand gestures, such as “Wash your hands”, (3) Point to things in a book when asked, (4) Follow two-step directions, such as “Get your shoes and put them in the basket”, and (5) Understand words such as “in”, “on”, and “under”. Response options to each of these items include *Yes* (coded as a 0) and *No* (coded as a 1). While these items were examined individually, items were also summed into a count measure that reflects the number of receptive language difficulties (ranging from 0 to 5).

*Expressive.* The remaining six items pertaining to early language development assess children’s expressive language skills. Specifically, primary caregivers were asked whether the child was able to do the following at the time of the survey: (1) Say at least one word, such as “hi” or “dog”, (2) Use two words together, such as “car go”, (3) Use three words together in a sentence, such as, “Mommy come now”, (4) Ask questions like “who”, “what”, “when”, “where”, (5) Ask questions like “why” and “how”, and (6) Tell a story with a beginning, middle, and end. Response options to each of these items include *Yes* (coded as a 0) and *No* (coded as a 1). While these items were examined individually, items were also summed into a count measure that reflects expressive language difficulties (ranging from 0 to 6).

*Composite.* Finally, a composite count measure of all 11 early language items was also included, which sums the number of receptive and expressive language difficulties (ranging from 0 to 11).

### 2.3. Independent Variable

A number of studies employing various iterations of the NSCH have examined households with children where one or more family members smoke tobacco products and have employed the term environmental tobacco smoke (ETS) to identify these households [48,49]. However, in line with more recent working using the NSCH [14], we made sure to compare households where a family member smokes to households where smoking occurs inside of the housing unit. We do this by constructing three groups derived from the following two questions: (1) Does anyone living in your household use cigarettes, cigars, or pipe tobacco? and if yes, (2) Does anyone smoke inside your home? Households where primary caregivers responded “no” to the first question (suggesting that the household is a non-smoking household) were assigned a value of 0 (referred to as *No Household Smoking*); households where primary caregivers responded “yes” to the first question, but “no” to the second question (suggesting that the household is a smoking household, but that smoking does not occur inside the housing unit) were assigned a value of 1 (referred to as *Household Smoking*); children of primary caregivers who responded “yes” to the first and the second question (suggesting that the household is a smoking household and that smoking occurs inside the housing unit) were assigned a value of 2 (referred to as *Smoking Inside Housing Unit*).

### 2.4. Covariates

All multivariate analyses include the following covariates to minimize omitted variable bias and the likelihood of spurious results: child age, child sex, child race (black, Hispanic, other, with white as reference category), low birth weight (child weighed less than 2500 g at birth), maternal age at birth, parent marital status, parent education [from less than high school (1) to college degree or higher (4)], parent immigrant status (immigrant = 1), and household poverty ratio (as a percentage of the Federal Poverty Level (FPL): 100–199%, 200–399%, 400+%, with below the poverty line as the reference category).

### 2.5. Analytic Plan

The analysis proceeded as follows. First, we calculated the descriptive statistics pertaining to the restricted sample of children ages 1 to 5, stratified by child sex. Second, negative binomial regression models, which regress each of the three language difficulties count measures on household smoking, ETS, and covariates, were estimated, first for the full sample and then for the male and female subsamples. Negative binomial regression was employed due to the properties and the distribution of the outcome count measures, which are zero-inflated and over dispersed. Coefficients are reported in the form of incidence ratio ratios (IRR) alongside 95% confidence intervals. Statistical significance was set at α = 0.05.

Models were further stratified by sex to examine whether the language development of male and female children is equally susceptible to any influence of environmental tobacco exposure. Given the results of these models, we estimated the predicted probability of each of the eleven individual early language difficulties among males, stratified by ETS categories, to examine whether any specific language difficulty was particularly relevant. Finally, ancillary analyses using more strategic comparisons (i.e., between male children exposed to ETS and male children in smoking households who have not been exposed to ETS) were also conducted as a means of examining the robustness of the findings.

## 3. Results

We first turn to the results of the descriptive analyses, which are displayed in Table 1. The findings indicate that the average rate of early language difficulties to be low—1.38/11 in the full sample. Among males, the rate seemed to be slightly higher at 1.55 (versus 1.19 among females). In terms of ETS, most children were not exposed to household smoking in any form regardless of sex (86.97% in the full sample). Still, of the remaining 13.03% of children exposed to household smoking, only a small portion was exposed to smoking inside the housing unit (7.29% of smoker households, ~1% of the full sample). The sample was 51.83% male. Covariates were largely similar across males and females, with the average age hovering at 3.04 years and white children making up the bulk of the sample (68.41%). About 8% of children were born low-birthweight, and the average maternal age at the time of the child’s birth was 30.23 years. Most children had married parents (76.72%), and only 12.09% had parents who were immigrants to the U.S. While the bulk of families exhibited household poverty ratios at 400+% of the FPL, 11.43% of families were under the poverty threshold (i.e., FPL below 100%).

Next, we turn to the results of the negative binomial regression models examining the association between ETS and early language difficulties. The findings, which are displayed in Table 2, reveal a general pattern in which smoking inside the housing unit, but not household smoking in general, was associated with significant increases in the rate of receptive (IRR = 1.85, CI = 1.01–3.43. *p* < 0.05), expressive (IRR = 1.58, CI = 1.21–2.07. *p* < 0.01), and composite language difficulties (IRR = 1.81, CI = 1.32–3.51. *p* < 0.01) among the full sample of children (net of covariates). Examination of the estimates pertaining to the covariates reveals that age (IRR = 0.40, CI = 0.39–0.41. *p* < 0.01), male (IRR = 1.39, CI = 1.30–1.48. *p* < 0.01), black (IRR = 1.30, CI = 1.13–1.50. *p* < 0.01), Hispanic (IRR = 1.20, CI = 1.09–1.34. *p* < 0.01), low birth weight (IRR = 1.42, CI = 1.27–1.59. *p* < 0.01), maternal age at birth (IRR = 1.01, CI = 1.00–1.02. *p* < 0.05), parent education (IRR = 0.93, CI = 0.89–0.98. *p* < 0.01), parent immigrant (IRR = 1.44, CI = 1.30–1.60. *p* < 0.01), and FPL 400%+ (IRR = 0.85, CI = 0.75–0.96. *p* < 0.01) were all significantly associated with composite language difficulties as well. Notably, after partitioning the sample into male and female subsamples, the significant findings pertaining to smoking inside the housing unit only emerged among male children. Specifically, among the male subsample, smoking inside the housing unit was associated with a 201% increase in the rate of early receptive language difficulties (CI = 1.34–6.79. *p* < 0.01), a 132% increase in the rate of early expressive language difficulties (CI = 1.63–3.29. *p* < 0.01), and a 182% increase in the rate of early composite language difficulties (CI = 1.83–4.35. *p* < 0.01). In the case of females, smoking inside the housing unit was not significantly associated with any of the early language measures. Furthermore, as was the case with the full sample, associations between household smoking (in the absence of smoking inside the housing unit) and measures of early language difficulties were consistently null among both males and females.

The significant findings pertaining to male children in Table 2 are illustrated in finer detail in Figure 1, which plots the predicted probability of each item assessing language difficulties among males by ETS categories, after setting covariates to their means to minimize confounding. To elaborate, by setting covariates to their mean values, these estimates held essential covariates constant, such as age, which in this figure was set to 3.04 years (i.e., the average age of male children in this sample). This allows for the illustration of the association between ETS categories and language difficulties for the average male child in the sample across covariates. After setting covariates to their means, smoking inside the housing unit was consistently and significantly associated with elevations in the predicted probability of each of the 11 items pertaining to language difficulties relative to no household smoking (all statistically significant differences were evaluated at the α = 0.05 level). To illustrate, compared to male children with no ETS in the home, those with ETS inside the housing unit were nearly five times as likely to be unable to follow verbal directions and more than six times as likely to be unable to point to things when asked. When it comes to expressive language, those with ETS inside the housing unit were more than five times as likely to be unable to use two words together and nearly four times as likely to be unable to ask why or how questions compared to male children with no ETS in the home. In contrast, there was little difference in the predicted probabilities of each of the language difficulties among male children across households where family members smoke (but not inside the housing unit) and non-smoking households. Finally, we checked the robustness of the findings displayed in Table 2 (and illustrated in Table 1) by changing the ETS reference group to “Household Smoking” instead of “No Household Smoking” among the male subsample. The findings, which are displayed in Table 3, remained virtually unchanged from those in Table 2, as the positive, significant association between smoking inside the housing unit and language difficulties among male children (regardless of subtype) was robust to whether the association was estimated in reference to non-smoking households or smoking households with no smoking inside the housing unit.

Supplemental models were performed that further adjusted the regression analyses for additional covariates, including parent mental health and positive parenting practices. Ultimately, these variables did not alter the findings and yielded very similar substantive results. As such, we opted to retain a more parsimonious model to preserve degrees of freedom. Results of supplemental models are available upon request.

## 4. Discussion

Exposure to ETS is a serious public health issue that can create challenges for child development, including mental, emotional, and behavioral development [12,13,14]. The current study aimed to expand upon existing knowledge in this area by examining the association between ETS and language difficulties among toddlers and preschool-aged children in the United States. The findings revealed that male (but not female) children in households with ETS—specifically smoking inside the housing unit—exhibited significant difficulties in both receptive and expressive components of language. This relationship held, moreover, across all eleven language items. Moreover, such difficulties remained among children in homes with indoor smoking whether they were compared to children in non-smoking households or children in smoking households without indoor smoking. This finding suggests that children’s likely exposure to ETS when family members smoke in the housing unit may be driving this relationship, above and beyond merely living in a household where a family member uses tobacco.

Notably, the general pattern of adverse consequences of ETS for language skill development only among male children is consistent with prior studies that have found boys are at higher risk of neurobehavioral disorders stemming from SHS exposure [18,50], which has been shown to be more likely when children live in homes where family members smoke indoors [5,7]. Overall, these results echo findings implicating ETS in executive functions [33], which are robust correlates of language development in children [34]. Even so, the present study employed a large, nationally representative sample, thereby expanding upon previous research using local samples and providing evidence of the harmful impact of ETS for early receptive and expressive language development, particularly among male children. As noted previously, researchers finding more male neurocognitive susceptibility to toxin exposure more broadly have hypothesized that oxidative stress and systemic inflammation may be the process by which toxin exposure differentially impacts neurodevelopment in male and female children, ultimately yielding a stronger effect in boys [41]. Beyond that, there may be several physiological and chemical differences between boys and girls in relation to neurodevelopment that extend to diverse learning and cognitive phenotypes [42]. In the case of sex-specific associations between ETS and language difficulties specifically, future research is needed to tease apart possible mechanisms, as the present data are cross-sectional.

Our results point to several implications for prevention efforts that can work to both minimize ETS in households with young children as well as bolster early language development among these children. Although rates of ETS in households with children have declined in recent years, they nonetheless remain high [51]. We propose that public health initiatives and programmatic efforts designed to reduce household smoking among vulnerable populations may improve health outcomes among children and their families. One fruitful avenue is through the implementation of smoke-free home interventions. Usually, such interventions aim to alter smoking in the home through behavioral change programming, which can include implementing aversive stimuli when smoking is detected, social reinforcements when smoking does not occur, as well as coaching individuals on ways to maintain a smoke-free home. For instance, one recent intervention on 298 families with a resident tobacco smoker and a child under age 14 utilized real-time punishment (mildly aversive lights and sounds), social reinforcement (praise), as well as brief one-on-one coaching sessions to discuss smoking cessation strategies. Findings demonstrated the program reduced the average level of airborne particles by 13.1% [52]. Kegler and colleagues [53] moreover tested the efficacy of a minimal intervention to create smoke-free homes in low-income households in Atlanta with participants recruited from a random sample of 2-1-1 callers. This mail and call-based intervention strategy focusing on promoting change through persuasion, role modeling, goal setting, environmental cues, and written and verbal reinforcement of actions to create a smoke-free home resulted in a 15-percentage point increase in the implementation of a full ban on smoking in the home at the 6-month follow-up period (relative to control groups).

Other recent research found benefits of smoke-free home interventions in households with children experiencing health challenges. Specifically, Nicholson and colleagues [54] found that a targeted multicomponent behavioral program delivered by trained counselors (in person and via phone calls) was successful at increasing home and full (i.e., home plus car) smoking ban adoption. Considering our findings linking ETS to language difficulties among male children, the implementation of this type of intervention may help to diminish early language disparities among male children. Additionally, one pediatric review found that household smoking restrictions were found to be effective in reducing childhood ETS by 20–50%, ultimately concluding that “childhood ETS is clearly reduced if smoke-free home policies are strictly implemented” [55].

Early childhood language skills may also be supported through the expansion of home visiting programs targeting vulnerable families. Home visiting programs aim to promote maternal and child well-being by providing information on child development and connecting at-risk families to critical resources (e.g., housing, education, employment, nutrition). Such policies can be specifically tailored to educate families on the harmful effects of smoking and ETS, as well as connect families where a household member smokes to resources, including tobacco cessation programs [56,57]. Finally, efforts can also be made toward providing specialized education to children exposed to ETS to enhance language skills. For instance, a randomized controlled trial of a 3-month caregiver implemented language development intervention using specific language facilitation strategies at home targeted for at-risk toddlers (up to 24 months old) was found to increase receptive and expressive language skills [58] significantly. Such programs can also be implemented in conjunction with schools, such as through programs such as the Getting Ready initiative. Specifically, this program is structured for teachers to provide support to parents (often through home visiting) to enhance the quality of parent-child learning experiences and interactions, as well as to create a system of shared responsibility between parents and educators for a child’s learning [59]. A randomized trial evaluation of the Getting Ready initiative among children ages 3–5 found significant improvements in children’s language, reading, and writing skills [60].

The current study has a few limitations that can be expanded upon in future research. First, because the NSCH is a cross-sectional survey, any findings of the current study should not be inferred to be necessarily causal. Second, the measure of ETS specifically refers to smoking inside the housing unit. Therefore, this item may miss the potential for children to be exposed to ETS in other spaces, such as automobiles. Still, to the extent that this measure may miss some children who were exposed to ETS in other settings, this would likely downwardly bias our estimates. Third, the measure of ETS is based on caregiver reports and therefore cannot directly assess SHS exposure. Our findings should be corroborated in future studies using alternative measures (e.g., serum cotinine levels). Fourth, the NSCH questionnaire does not explicitly ask about ETS emanating from electronic cigarettes. It would be useful for future research to explore whether ETS from electronic cigarettes is harmful for children’s language acquisition. Fifth, some potential confounders during the prenatal period (e.g., smoking during pregnancy) are not available in the NSCH data. Even so, we were able to account for low birth weight, which has been linked to both gestational tobacco exposure [61,62] and developmental learning deficits [63,64]. Relatedly, the current data also lacked information on potentially important measures such as infant nutrition and diet, maternal health, genetic factors, and children’s level of socialization and parent-child scholarization. Future research—especially those with longitudinal data—should consider the role of these factors in the association between environmental tobacco smoke exposure and early language development. Finally, the measure of early language skills also relied on caregiver reports. Thus, it is possible that caregivers may misjudge the language skills of children. Even so, research has suggested that such reports are generally consistent with direct assessment [65,66] and that both are similarly predictive of language delay at age 3 [67]. Nevertheless, it would be useful for future research to replicate the present findings using direct, individual language assessment.

## 5. Conclusions

The findings from this study bolster the need for greater awareness and prevention efforts to reduce ETS among young children. We proposed that programmatic efforts designed to reduce ETS in households with young children, as well as interventions that target early language development among at-risk children, may be beneficial to enhance child development and well-being. To maximize this effort, interdisciplinary teams, including medical and public health practitioners, educators, and researchers, can work together to develop and implement evidence-based strategies to improve the health and development of children. In conjunction with the findings from a growing body of research linking both ETS and SHS to adverse child outcomes, our results may help to inform decisions of policymakers in their efforts to minimize ETS in households with children and to guide future prevention efforts.

## Figures and Tables

**Figure 1 ijerph-18-06489-f001:**
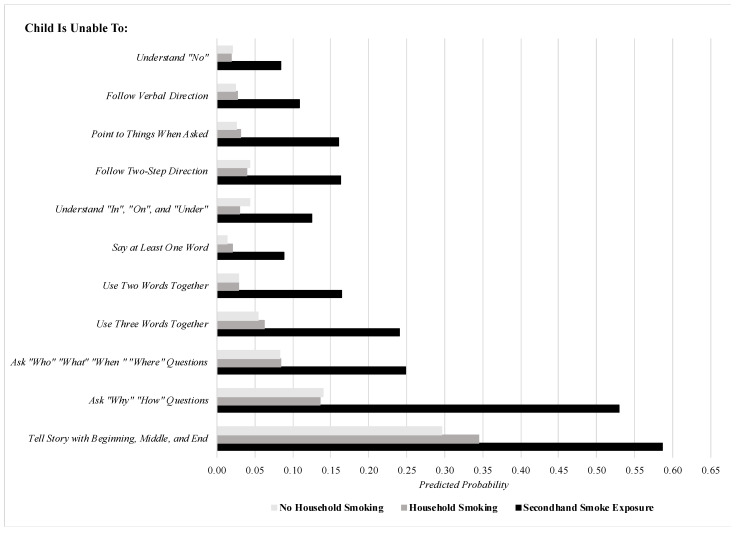
The predicted probability of specific early language difficulties among males, stratified by ETS categories (covariates set to means). Note: predicted probabilities comparing smoking inside housing unit to no household smoking are all statistically significant at the *p* < 0.05 level.

**Table 1 ijerph-18-06489-t001:** Descriptive statistics pertaining to children aged 1 to 5, stratified by child sex.

Variables	Full Sample (*N* = 7594)			Males (*N* = 3936)			Females (*N* = 3658)		
	Mean/%	SD	Range	Mean/%	SD	Range	Mean/%	SD	Range
*Early Language Difficulties*									
Receptive	0.31	0.90	0–5	0.35	0.96	0–5	0.26	0.83	0–5
Expressive	1.07	1.68	0–6	1.20	1.76	0–6	0.93	1.57	0–6
Composite	1.38	2.36	0–11	1.55	2.50	0–11	1.19	2.19	0–11
*Environmental Tobacco Smoke (ETS)*									
No Household Smoking	86.97%	0.34	0–1	86.92%	0.34	0–1	87.04%	0.34	0–1
Household Smoking	12.08%	0.33	0–1	12.26%	0.33	0–1	11.86%	0.32	0–1
Smoking Inside Housing Unit	0.95%	0.10	0–1	0.82%	0.09	0–1	1.09%	0.10	0–1
*Covariates*									
Age	3.04	1.37	1–5	3.04	1.38	1–5	3.05	1.37	1–5
Male	51.83%	0.50	0–1	-	-	-	-	-	-
White	68.41%	0.46	0–1	68.60%	0.46	0–1	68.21%	0.47	0–1
Black	5.69%	0.23	0–1	5.56%	0.23	0–1	5.82%	0.23	0–1
Hispanic	11.85%	0.32	0–1	11.59%	0.32	0–1	12.14%	0.33	0–1
Other Race/Ethnicity	14.05%	0.35	0–1	14.25%	0.35	0–1	13.83%	0.35	0–1
Low Birth Weight	7.99%	0.27	0–1	7.44%	0.26	0–1	8.58%	0.28	0–1
Maternal Age at Birth	30.23	5.40	18–45	30.26	5.44	18–45	30.19	5.35	18–45
Parental Marital Status	76.72%	0.42	0–1	76.93%	0.42	0–1	76.49%	0.42	0–1
Parent Education	3.49	0.77	1–4	3.50	0.76	1–4	3.48	0.77	1–4
Parent Immigrant	12.09%	0.33	0–1	12.22%	0.33	0–1	11.95%	0.32	0–1
FPL Below 100%	11.43%	0.32	0–1	11.28%	0.32	0–1	11.59%	0.32	0–1
FPL 100–199%	16.29%	0.37	0–1	16.95%	0.38	0–1	15.58%	0.36	0–1
FPL 200–399%	31.56%	0.46	0–1	31.91%	0.47	0–1	31.19%	0.46	0–1
FPL 400%+	40.72%	0.49	0–1	39.86%	0.49	0–1	41.63%	0.49	0–1

**Table 2 ijerph-18-06489-t002:** Negative binomial regression of the association between environmental tobacco smoke (ETS) and early language difficulties.

	Full Sample			Males			Females		
	*Early Language Difficulties*			*Early Language Difficulties*			*Early Language Difficulties*		
	*Receptive*	*Expressive*	*Composite*	*Receptive*	*Expressive*	*Composite*	*Receptive*	*Expressive*	*Composite*
Variables	IRR	IRR	IRR	IRR	IRR	IRR	IRR	IRR	IRR
	(95% CI)	(95% CI)	(95% CI)	(95% CI)	(95% CI)	(95% CI)	(95% CI)	(95% CI)	(95% CI)
*Environmental Tobacco Smoke (ETS)*									
Household Smoking	0.97	1.04	1.01	0.91	1.05	1	1.08	1.01	1.01
	(0.79–1.20)	(0.95–1.13)	(0.91–1.12)	(0.69–1.20)	(0.94–1.17)	(0.88–1.15)	(0.78–1.49)	(0.89–1.15)	(0.87–1.18)
Smoking Inside Housing Unit	1.85 *	1.58 **	1.81 **	3.01 **	2.32 **	2.82 **	0.6	1	0.89
	(1.01–3.43)	(1.21–2.07)	(1.32–1.51)	(1.34–6.79)	(1.63–3.29)	(1.83–4.35)	(0.20–1.80)	(0.65–1.54)	(0.54–1.46)
*Covariates*									
Age	0.43 **	0.38 **	0.40 **	0.44 **	0.41 **	0.42 **	0.40 **	0.34 **	0.37 **
	(0.41–0.45)	(0.37–0.39)	(0.39–0.41)	(0.41–0.47)	(0.39–0.42)	(0.41–0.44)	(0.37–0.44)	(0.33–0.36)	(0.35–0.38)
Male	1.38 **	1.32 **	1.39 **	-	-	-	-	-	-
	(1.21–1.57)	(1.25–1.39)	(1.30–1.48)						
Black	1.55 **	1.11	1.30 **	2.05 **	1.13	1.45 **	0.96	1.07	1.07
	(1.17–2.04)	(0.99–1.25)	(1.13–1.50)	(1.41–2.98)	(0.97–1.31)	(1.20–1.77)	(0.63–1.48)	(0.90–1.26)	(0.87–1.32)
Hispanic	1.2	1.12 **	1.20 **	1.26	1.14 *	1.24 **	1.13	1.07	1.14
	(0.97–1.49)	(1.03–1.21)	(1.09–1.34)	(0.94–1.68)	(1.02–1.27)	(1.08–1.43)	(0.82–1.58)	(0.95–1.21)	(0.98–1.33)
Other Race/Ethnicity	1.02	1.05	1.07	1.15	1.02	1.07	0.87	1.1	1.07
	(0.83–1.25)	(0.98–1.14)	(0.97–1.18)	(0.88–1.49)	(0.92–1.13)	(0.94–1.22)	(0.63–1.20)	(0.98–1.24)	(0.92–1.24)
Low Birth Weight	1.59 **	1.28 **	1.42 **	1.38 *	1.22 **	1.30 **	1.92 **	1.35 **	1.58 **
	(1.26–2.00)	(1.17–1.40)	(1.27–1.59)	(1.02–1.88)	(1.08–1.38)	(1.11–1.52)	(1.35–2.72)	(1.18–1.53)	(1.34–1.87)
Maternal Age at Birth	1.01	1.01 *	1.01 *	1.02	1.01 *	1.01 **	1	1.01	1
	(0.99–1.02)	(1.00–1.02)	(1.00–1.02)	(1.00–1.03)	(1.00–1.01)	(1.00–1.02)	(0.98–1.02)	(1.00–1.01)	(0.99–1.01)
Parental Marital Status	0.83 *	1	0.93	0.89	0.95	0.9	0.75 *	1.06	0.96
	(0.69–0.99)	(0.93–1.07)	(0.85–1.02)	(0.70–1.14)	(0.87–1.05)	(0.79–1.01)	(0.57–0.99)	(0.95–1.18)	(0.84–1.10)
Parent Education	0.91	0.97	0.93 **	0.82 **	0.95	0.89 **	1.04	0.98	0.98
	(0.82–1.01)	(0.93–1.01)	(0.89–0.98)	(0.72–0.94)	(0.90–1.01)	(0.84–0.96)	(0.89–1.21)	(0.92–1.04)	(0.91–1.05)
Parent Immigrant	1.66 **	1.27 **	1.44 **	1.57 **	1.28 **	1.42 **	1.86 **	1.24 **	1.48 **
	(1.34–2.06)	(1.17–1.37)	(1.30–1.60)	(1.19–2.07)	(1.15–1.42)	(1.23–1.63)	(1.33–2.60)	(1.10–1.40)	(1.26–1.73)
FPL 100–199%	0.87	0.99	0.95	0.92	0.99	0.98	0.82	0.98	0.92
	(0.69–1.11)	(0.90–1.09)	(0.85–1.08)	(0.66–1.26)	(0.87–1.13)	(0.83–1.15)	(0.57–1.19)	(0.85–1.13)	(0.77–1.10)
FPL 200–399%	0.88	0.97	0.94	0.95	0.97	0.97	0.78	0.97	0.9
	(0.70–1.11)	(0.88–1.06)	(0.83–1.05)	(0.69–1.30)	(0.85–1.09)	(0.83–1.14)	(0.55–1.11)	(0.85–1.12)	(0.76–1.07)
FPL 400%+	0.68 **	0.92	0.85 **	0.74	0.92	0.87	0.61 **	0.92	0.82 *
	(0.53–0.88)	(0.83–1.01)	(0.75–0.96)	(0.53–1.03)	(0.81–1.05)	(0.74–1.02)	(0.42–0.88)	(0.80–1.06)	(0.69–0.99)
*N*	7594	7594	7594	3936	3936	3936	3658	3658	3658

Abbreviations: IRR = incidence ratio ratio; CI = confidence interval. ** *p* < 0.01, * *p* < 0.05.

**Table 3 ijerph-18-06489-t003:** Strategic comparison of early language difficulties between ETS categories among the male sample.

	*Early Language Difficulties*		
	*Receptive*	*Expressive*	*Composite*
Environmental Tobacco Smoke (ETS)	IRR (95% CI)	IRR (95% CI)	IRR (95% CI)
*Strategic Comparison of Groups*:			
Smoking Inside Housing Unit v. No Household Smoking	3.01 ** (1.34–6.79)	2.32 ** (1.63–3.29)	2.82 ** (1.83–4.35)
Smoking Inside Housing Unit v. Household Smoking	3.31 ** (1.43–7.76)	2.20 ** (1.54–3.16)	2.81 ** (1.80–4.39)
*N*	3936	3936	3936

Abbreviations: IRR = incidence ratio ratio; CI = confidence interval. ** *p* < 0.01. All covariates from Table 2 included, but suppressed to conserve space.

## Data Availability

The data employed for the present study are publicly available through the Child and Adolescent Health Measurement Initiative.

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
