# Peer review of "Environmental Tobacco Smoke and Early Language Difficulties among U.S. Children"

_ijerph, 2021, doi:10.3390/ijerph18126489_

Round 1
Reviewer 1 Report
This study assessed the association between the environmental tobacco smoke and language difficulties among children aged 1-5 years and found the disparities among males and females. Overall, this paper was written well but a few things need to think about and could be improved.
- For introduction, why authors using the citation of USDHHS 2014 instead of USDHHS 2016 (the relevant recent evidence regarding second hand smoking)?
- The introduction for ETS is prolix and for sex-difference is too brief. Recommend authors to rewrite partly to balance the content.
- For methods part, authors should state their statistical part elaborately, such as what is IRR (incidence rate ratio) 95% CI, significance level, etc. At least they should appear in the table footnote to let the readers know.
- Although authors tried to include the covariates/confounders for adjusting the models, other factors that they may need to think about: maternal/paternal mental health, infant nutrition/diet, maternal BMI, genetic factors and so on.
- For results part, I would recommend to add another group No Household Smoking vs. Household Smoking in table 3 since from the IRR, we can tell the results are quite similar and we would like to know the differences between the other groups.
- The authors said "The significant findings pertaining to male children in Table 2 are illustrated in finer detail in Figure 1. So did they compare the three groups in figure 1? If so, the significance should be marked in figure 1 such as (p-values).
- For the discussion, authors listed the previous evidence that found regarding the relationships among males. However, it lack of the explanation for the mechanisms. Why is this? Is it due to the different development rates between male and female children? Or other culture/social differences?
Author Response
This study assessed the association between the environmental tobacco smoke and language difficulties among children aged 1-5 years and found the disparities among males and females. Overall, this paper was written well but a few things need to think about and could be improved.
- For introduction, why authors using the citation of USDHHS 2014 instead of USDHHS 2016 (the relevant recent evidence regarding second hand smoking)?
Response: Thank you for pointing us to this work. While we have retained the original citation, since that work specifically supports to cost estimates that we cited, we have added newer citations to include more updated statistics about the consequences of secondhand smoke exposure (Tsai et al., 2018; U.S. Department of Health and Human Services 2020; Yousuf et al., 2020).
- The introduction for ETS is prolix and for sex-difference is too brief. Recommend authors to rewrite partly to balance the content.
Response: The introduction has now been revised to expand the discussion of sex-differences in order to balance the content.
- For methods part, authors should state their statistical part elaborately, such as what is IRR (incidence rate ratio) 95% CI, significance level, etc. At least they should appear in the table footnote to let the readers know.
Response: We have now more clearly defined incidence ratio, 95% confidence intervals and statistical significance levels both in the text and in the tables.
- Although authors tried to include the covariates/confounders for adjusting the models, other factors that they may need to think about: maternal/paternal mental health, infant nutrition/diet, maternal BMI, genetic factors and so on.
Response: Thank you for the helpful suggestion. We have adjusted the model based on the relevant characteristics captured in the NSCH data. However, many of these other potential relevant factors are either not captured in the survey data (i.e., genetic factors; infant nutrition) or if they are captured (parent mental health) the temporal ordering is not antecedent to both the focal independent and dependent variable. Still, we have run supplemental analysis including a measure of parent mental health as a covariate, as well as positive parenting practices (as suggested by another reviewer) and the substantive findings do not change. The results of these supplemental models are discussed in the last paragraph of the results section. Ultimately, we have decided to retain our more parsimonious model, but have now noted in the limitations section that future research to consider the role of these factors.
- For results part, I would recommend to add another group No Household Smoking vs. Household Smoking in table 3 since from the IRR, we can tell the results are quite similar and we would like to know the differences between the other groups.
Response: Thank you for your comment. The estimate being requested (comparing no household smoking to household smoking) is already presented in Table 2 (see the very first column showing household smoking has no impact on language difficulties unless someone is smoking inside the housing unit). Additionally, comparisons to the no household smoking group do not constitute a strategic comparison between two more comparable groups (households where someone smokes and households where someone smokes inside the housing unit).
- The authors said "The significant findings pertaining to male children in Table 2 are illustrated in finer detail in Figure 1. So did they compare the three groups in figure 1? If so, the significance should be marked in figure 1 such as (p-values).
Response: We have now added a footnote to table 1, as well as a statement in the results section of the text that all predicted probabilities comparing smoking inside housing unit to no household smoking are statistically significant at the p <.05 level.
- For the discussion, authors listed the previous evidence that found regarding the relationships among males. However, it lack of the explanation for the mechanisms. Why is this? Is it due to the different development rates between male and female children? Or other culture/social differences?
Response: Thank you for this helpful suggestion. In line with another request from another reviewer to expound upon potential explanations of male-female differences up front (justifying the expectation), we have added this text mostly to the introduction and now briefly revisit it in the discussion. Importantly, the current study cannot ascertain the exact mechanisms due to the cross-sectional nature of the data. Still, we call for future research to further assess these possibilities.
Reviewer 2 Report
Even if the approach of the study is more qualitative than quantitative, both positive aspects and limitations in the results are well described.
Only one important item is not taken into account: the level of socialization or scholarization of the kids (Does the baby go to daycare/Kinder garden/preschool?).
Please insert a comment about it. If the informations about scholarization of the kids is known, please insert these data in the multiple regession analisys, otherwise insert the lack of informations in the limitations of the study.
Author Response
- Even if the approach of the study is more qualitative than quantitative, both positive aspects and limitations in the results are well described.
Response: Thank you very much for the positive assessment of the manuscript and the helpful suggestions.
- Only one important item is not taken into account: the level of socialization or scholarization of the kids (Does the baby go to daycare/Kinder garden/preschool?).
Response: Thank you for this helpful suggestion. Unfortunately, the data lack information on the level of socialization/scholarization of the kids. However, we have now reported the results of a supplemental analysis in the last paragraph of the results section noting that that the results remain substantively identical even after including additional covariates for parental mental health and positive parenting practices. We have noted in the limitations section the need for future research to consider the role of socialization and scholarization in the relationship between environmental tobacco smoke exposure and early learning skills.
- Please insert a comment about it. If the informations about scholarization of the kids is known, please insert these data in the multiple regession analisys, otherwise insert the lack of informations in the limitations of the study.
Response: We have now noted in the manuscript in the last paragraph of the results section that additional analyses were performed that further adjusted for additional covariates including parent mental health and positive parenting practices. Ultimately, these variables did not alter the findings and yielded very similar substantive results. As such, we have opted to retain a more parsimonious models to preserve degrees of freedom. However, as noted in the comment above, this has now been added as a potential limitation in the revised manuscript.
Reviewer 3 Report
This is a very interesting investigation with robust results related to association between family members smoking inside the housing unit and both receptive and expressive language difficulties.
The findings reveal a robust association between family members smoking inside the housing unit and both receptive and expressive language difficulties, but only among 22 male children. Nevertheless, the authors must formulate a hypothesis explaining why they do not have the same results for women, and investigate whether in any of the studies that they mentioned about the affectation of tobacco, there were any results that show differences between male and females.
Author Response
- This is a very interesting investigation with robust results related to association between family members smoking inside the housing unit and both receptive and expressive language difficulties.
Response: Thank you for the positive assessment of the manuscript and helpful comments.
- The findings reveal a robust association between family members smoking inside the housing unit and both receptive and expressive language difficulties, but only among 22 male children. Nevertheless, the authors must formulate a hypothesis explaining why they do not have the same results for women, and investigate whether in any of the studies that they mentioned about the affectation of tobacco, there were any results that show differences between male and females.
Response: Thank you for the helpful comment. We have now laid out this hypothesis more clearly in the introduction and revisit it in the discussion as well. Thank you for pointing this out.
Round 2
Reviewer 1 Report
The authors have provided substantial edits in response to the last round of comments. The literature review is adequate, as is the discussion. Their method is more clearly and accurately described. In my opinion, the piece is ready for publication.